# Butyrate and other short-chain fatty acids increase the rate of lipolysis in 3T3-L1 adipocytes

John M. Rumberger[1,2], Jonathan R.S. Arch[2] and Allan Green[3]

[1] Bassett Healthcare, Cooperstown, NY, USA
[2] Clore Laboratory, University of Buckingham, Buckingham, UK
[3] Department of Chemistry and Biochemistry, SUNY Oneonta, Oneonta, NY, USA

## ABSTRACT

We determined the effect of butyrate and other short-chain fatty acids (SCFA) on rates of lipolysis in 3T3-L1 adipocytes. Prolonged treatment with butyrate (5 mM) increased the rate of lipolysis approximately 2–3-fold. Aminobutyric acid and acetate had little or no effect on lipolysis, however propionate stimulated lipolysis, suggesting that butyrate and propionate act through their shared activity as histone deacetylase (HDAC) inhibitors. Consistent with this, the HDAC inhibitor trichostatin A (1 μM) also stimulated lipolysis to a similar extent as did butyrate. Western blot data suggested that neither mitogen-activated protein kinase (MAPK) activation nor perilipin down-regulation are necessary for SCFA-induced lipolysis. Stimulation of lipolysis with butyrate and trichostatin A was glucose-dependent. Changes in AMP-activated protein kinase (AMPK) phosphorylation mediated by glucose were independent of changes in rates of lipolysis. The glycolytic inhibitor iodoacetate prevented both butyrate- and tumor necrosis factor-alpha-(TNF-$\alpha$) mediated increases in rates of lipolysis indicating glucose metabolism is required. However, unlike TNF-$\alpha$−, butyrate-stimulated lipolysis was not associated with increased lactate release or inhibited by activation of pyruvate dehydrogenase (PDH) with dichloroacetate. These data demonstrate an important relationship between lipolytic activity and reported HDAC inhibitory activity of butyrate, other short-chain fatty acids and trichostatin A. Given that HDAC inhibitors are presently being evaluated for the treatment of diabetes and other disorders, more work will be essential to determine if these effects on lipolysis are due to inhibition of HDAC.

## INTRODUCTION

High rates of adipose tissue lipolysis can lead to production of excess free fatty acids. Excess free fatty acids increase the rate of hepatic glucose output, induce skeletal muscle insulin resistance, and have other adverse effects that contribute to development of diabetes and cardiovascular disease (*Bergman & Ader, 2000*; *Ginsberg, 2000*; *Egan, Greene & Goodfriend, 2001*). Much is known about how adipose tissue lipolysis is regulated acutely or minute-to-minute but little is known about long-term regulation over time periods

Corresponding author
Allan Green,
allan.green@oneonta.edu

relevant to the progression of chronic disease. A better understanding of the mechanisms regulating rates of lipolysis over the long term may reveal new targets for therapeutic intervention (*Bergman & Mittelman, 1998*; *Large & Arner, 1998*; *Green, 2006*).

Significant quantities of short-chain fatty acids (SCFA) are produced through fermentation of dietary fibers in the lower intestinal tract. In humans SCFA constitute approximately 10% of the caloric energy absorbed (*Bergman, 1990*). Near millimolar concentrations of butyrate are found in the hepatic portal vein, and concentrations *in vivo* may be physiologically significant for the regulation of adipocyte $\beta$-adrenergic receptor gene expression (*Bergman, 1990*; *Krief et al., 1994*). It has been reported that SCFA influence lipid metabolism, $\beta$-adrenergic receptor concentrations, and leptin production (*Krief et al., 1994*; *Ding et al., 2000*; *Metz, Lopes-Cardoza & Van den Berg, 1974*; *Xiong et al., 2004*).

SCFA have a number of effects on cells, many of which, especially those of butyrate, are mediated through inhibition of histone deacetylases (HDACs) (*Waldecker et al., 2008*; *Marshall et al., 2003*; *Kruh, 1982*). HDACs are involved in the pathogenesis of diabetes and are currently of interest as targets for the treatment of several diseases including diabetes and cancer (*Das & Kundu, 2005*; *Gray & De Meyts, 2005*; *Christensen et al., 2011*). In addition, SCFA have been shown to be ligands for the orphan G protein-coupled receptors (GPCR) GPR41 and GPR43 (*Brown et al., 2003*). GPR41 has been reported to mediate the effects of SCFA on leptin production in adipocytes (*Xiong et al., 2004*).

Given that both GPCR and HDACs are under active investigation as therapeutic targets for a wide spectrum of diseases, we conducted this study to determine whether SCFA affect rates of lipolysis in adipocytes.

## METHODS AND MATERIALS

### Materials

3T3-L1 cells (ATCC CL-173) were obtained from ATCC (Manassas, VA); glucose-containing DMEM and antibiotics were from Atlanta Biologicals (Norcross, GA); glucose-free DMEM was from Irvine Scientific (Santa Ana, CA); fetal bovine serum was from Hyclone Laboratories, Inc. (Logan, UT); insulin (Humulin® R7) was from Eli Lilly and Co. (Indianapolis, IN); BSA was from Intergen Co. (Purchase, NY); I-Block was from Pierce (Rockford, IL); glutamine was from Gibco (Grand Island, NY); anti-ERK 1/2 and anti-active MAP kinase antibodies were from Promega (Madison, WI); anti-AMPK$\alpha$ and anti-phospho-AMPK$\alpha$ (T172) antibodies were from Cell Signaling Technology (Beverly, MA); secondary antibody (donkey anti-rabbit HRP conjugate) was from Santa Cruz Biotechnology (Santa Cruz, CA); and glycerol reagent for glycerol release assay was from Amresco (Solon, OH) and lactate assay reagent was from Trinity Biotech (St. Louis, MO). All other reagents were from Sigma (St. Louis, MO). Short-chain fatty acids were purchased as sodium salts and dissolved in DMEM.

### Cell culture

3T3-L1 cells were cultured in 24-well plates and maintained as previously described (*Green et al., 2004*) in standard medium (DMEM with high glucose, supplemented with 10%

fetal bovine serum and with PSA (penicillin 100 units/ml, streptomycin 100 μg/ml, and amphotericin 0.25 μg/ml). Medium was changed every 2–3 days. At 2–4 days after confluence, differentiation into adipocytes was initiated as follows: standard medium was supplemented with 5 μg/ml insulin, 0.5 μg/ml dexamethasone, and 0.5 mmol/l 3-isobutyl-1-methylxanthine for 2 days. The medium was then changed and supplemented with insulin only for 2–3 days. Thereafter, the cells were maintained in standard medium only. Cells were used 3–10 days post-differentiation. For experimental conditions without glucose, cells were incubated in DMEM without glucose, supplemented with 1% BSA, 4 mM glutamine, 44 mM $NaHCO_3$, 20 mM HEPES and 0.01% pyruvic acid.

### Glycerol assay

Lipolysis was measured as the rate of glycerol release, as previously described (*Green et al., 2004*). After the various treatments, cells were washed three times with DMEM, and then incubated for another 1 h. Media were then collected from the cells and heated at 65 °C for 8 min to inactivate any enzymes released from the cells. Samples (50 μl) were then assayed for glycerol using 150 μl glycerol reagent in a flat bottom 96-well plate. Absorption was measured at 500 nm on a Molecular Devices plate reader.

### Lactate assay

Lactate concentrations were determined colorimetrically, using a kit from Trinity Bioech (St. Louis, MO), by following instructions provided by the manufacturer.

### Western blots

Western immunoblots were performed by slight modifications of our previously reported methods (*Green et al., 2004*; *Gasic, Tian & Green, 1999*), as follows. Cells were harvested in Laemmli sample buffer (*Laemmli, 1970*) and aspirated with a syringe five times through a 25 g needle. The samples were centrifuged (16,000 g, 30 s) to remove fat, and then heated at 95 °C for 5 min prior to being resolved on SDS polyacrylamide gels (10%). Proteins were transferred to nitrocellulose membranes. Membranes were blocked with 5% blotto, 1% BSA or 0.2% I-Block, and probed with polyclonal rabbit antibodies raised against perilipin (gift of Dr. Andrew Greenberg, Human Nutrition Research Center, Tufts University, Boston, MA, USA), AMPK$\alpha$, phospho-AMPK$\alpha$ (T172), ERK1/2 or active MAP Kinase (each at dilutions of 1:3,000). After incubation with anti-rabbit IgG-HRP (diluted 1:10,000), the blots were developed with ECL Plus and visualized with Hyperfilm ECL (Amersham Pharmacia Biotech, Piscataway, NJ).

### Statistics

Differences between pairs of treatments were analyzed by Student's *t*-test. A *P*-value of less than 0.05 was considered statistically significant. *P*-values are given in the figure legends.

## RESULTS

To investigate the effect of butyrate on lipolysis, 3T3-L1 adipocytes were incubated for up to 4 h with 5 mM butyrate then washed, and the rate of glycerol release was determined

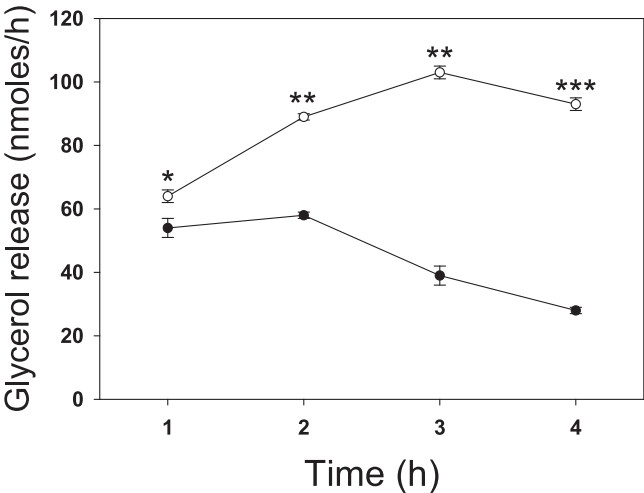

**Figure 1 Time dependence of butyrate on rates of lipolysis in 3T3-L1 adipocytes.** 3T3 L1 adipocytes were incubated with nothing (●-●) or 5 mM butyrate (○-○). At the indicated times cells were washed and incubated for one hour, and then glycerol was measured as an index of the rate of lipolysis, as described in the methods section. Data depicted are representative of three or more independent experiments. Data shown are means ± SE ($n = 3$). *$P < 0.05$; **$P < 0.01$; ***$P < 0.001$.

over the next hour (Fig. 1). The rate of lipolysis initially increased in both control and treated cells over the first hour, but to a similar degree. At 1 h treatment with butyrate the rate of lipolysis was similar to that of control cells with only a slight (but statistically significant) stimulatory effect. However, there was then a time-dependent marked increase in the rate of lipolysis in butyrate-treated cells, with maximal stimulation occurring between 3 and 4 h (Fig. 1). Maximal rates of lipolysis with butyrate were similar in time course studies for at least 48 h (data not shown).

We next investigated the effect of a series of related SCFA (Fig. 2). Similar to the 4 h studies, 5 mM butyrate caused a 2–3-fold increase in the rate of lipolysis whereas 5 mM acetate or 2-aminobutyric acid had little or no effect. We hypothesized that the known HDAC inhibitory activity of butyrate underlies its lipolytic effect, and so we evaluated the effects of 20 mM propionate, a less potent HDAC inhibitor than butyrate, and 1 μM trichostatin A, a potent and specific small molecule HDAC inhibitor. Both propionate and trichostatin A increased rates of lipolysis to a similar extent as did butyrate. Together these findings suggest that HDAC inhibition is involved in the lipolytic effect of these compounds.

As we have used SCFA at concentrations at which HDAC inhibitory activity is maximal (*Marshall et al., 2003*), the possibility remains that lower concentrations of SCFA also increase rates of lipolysis, which would imply action through an alternative mechanism. Figure 3 shows the dose-dependence of the various SCFA on rates of lipolysis. The half-maximal concentrations of butyrate and propionate were in the low millimolar range, and the relative potencies of the SCFA were butyrate > propionate > acetate. Both the absolute and the relative potencies of the SCFA indicate that inhibition of HDAC is important for increasing rates of lipolysis, and that it is unlikely they are having this effect through activation of a G protein-coupled receptor.

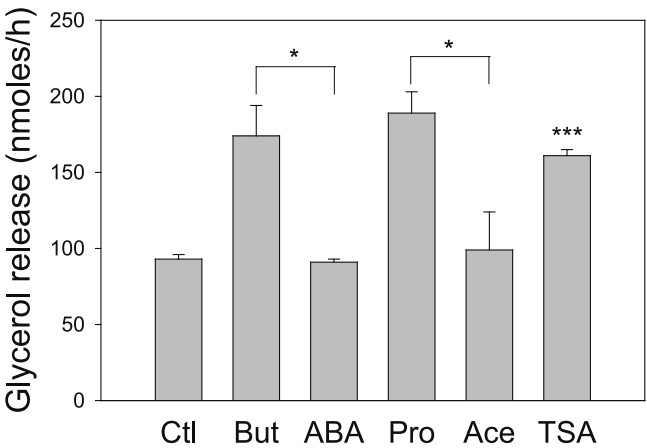

**Figure 2 Effect of Short-Chain fatty acids and trichostatin A on lipolysis.** 3T3-L1 adipocytes were treated for 18 h with no additions (Ctl); 5 mM Butyrate (But), 5 mM 2-amino-butyrate (ABA), 5 mM acetate (Ace); 20 mM propionate (Pro); or 1 μM Trichostatin A (TSA). Rates of glycerol release were then determined as in the legend for Fig. 1. Data depicted are representative of three or more independent experiments. Data shown are means ± SE ($n = 3$). *$P < 0.05$; ***$P < 0.001$ (Trichostatin A compared to control).

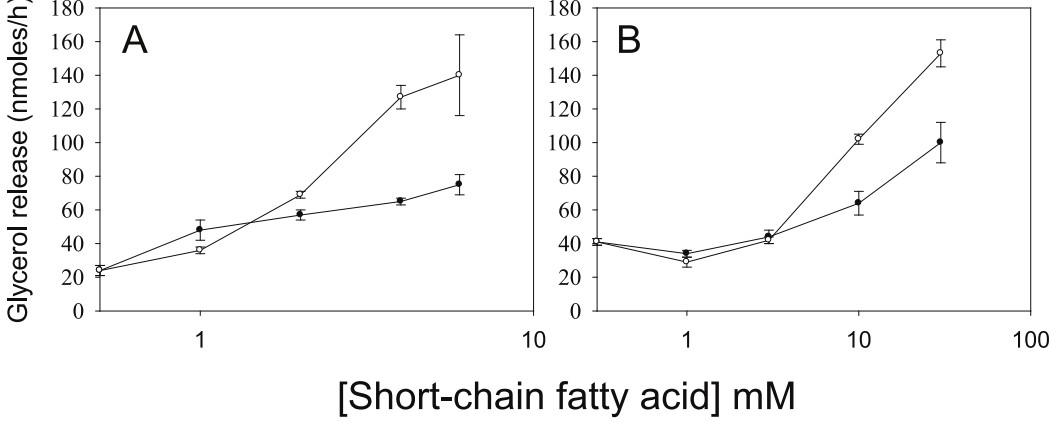

**Figure 3 Relative lipolytic potencies of Short-Chain fatty acids.** 3T3-L1 adipocytes were treated for 24 h with the indicated concentrations of Short-Chain fatty acids. After incubation, cells were washed and rates of lipolysis determined as in the legend for Fig. 1. (A) Comparison of butyrate (○–○) with ABA (●–●). The concentrations were 0, 1, 2, 3 and 5 mM. (B) Comparison of propionate (○–○) with acetate (●-●). Concentrations were 0, 1, 3, 10 and 30 mM. Data depicted are representative of three or more independent experiments. Data shown are means ± SE ($n = 3$).

Butyrate has been shown to alter $\beta$-adrenergic receptor profiles in adipocytes (*Krief et al., 1994*; *Ding et al., 2000*). Therefore, we investigated whether this altered $\beta$-receptor profile might account for the effect of butyrate on rates of lipolysis. Assuming that $\beta$-adrenergic receptors have some constitutive activity in the absence of agonist (*Chidiac et al., 1994*), greater concentrations of $\beta$-adrenergic receptors would be expected to increase rates of lipolysis by increasing cellular concentrations of cyclic AMP. The increased concentration of cyclic AMP would in turn activate PKA and hence increase rates of

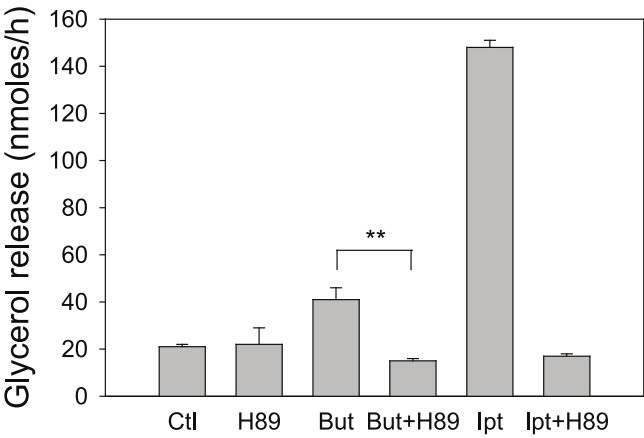

**Figure 4 Effect of the PKA inhibitor H-89 on rates of lipolysis.** 3T3-L1 adipocytes were treated for 4 h with glucose plus no further additions (Ctl) or 5 mM butyrate (But). Isoproterenol (1 μM) and H-89 (50 μM) were added as indicated and glycerol was measured after another 30 min. Data depicted are representative of three or more independent experiments. Data shown are means ± SE ($n = 3$). **$P < 0.01$.

lipolysis. To investigate this possibility we used the PKA inhibitor H-89, which would be expected to prevent the increase in lipolysis with butyrate treatment if the mechanism involves increases in cyclic AMP concentrations. Figure 4 depicts an experiment where 3T3-L1 adipocytes were treated for 4 h with glucose, with or without 5 mM butyrate. The PKA inhibitor H-89 (50 μM) was added for the last 30 min of the incubation, cells were washed, and rates of lipolysis were measured. H-89 prevented the stimulation of lipolysis by isoproterenol, the classic $\beta$-adrenergic receptor agonist. Similarly H-89 prevented the increased rates of lipolysis seen with butyrate suggesting that increased cyclic AMP concentrations underlie the butyrate effect on lipolysis.

Long term regulation of lipolysis by other mediators, such as TNF-$\alpha$, is thought to involve activation of the MAP kinases ERK1&2 and the down-regulation of perilipin (*Rosenstock, Greenberg & Rudich, 2001*; *Souza et al., 1998*; *Souza et al., 2003*; *Gronning et al., 2002*). To see whether these cellular events are important for butyrate-stimulated lipolysis we treated cells with or without butyrate for 18 h and then performed Western blots on total cell extracts for these proteins (Fig. 5). Although TNF-$\alpha$ treatment resulted in increased lipolysis along with activation of MAP kinase (shown by phosphorylation of ERK 1/2) and down-regulation of perilipin, butyrate treatment had no such effect. These data demonstrate that neither MAP kinase activation nor perilipin down-regulation is necessary for increasing rates of lipolysis in 3T3-L1 adipocytes. This was of interest because we have previously reported that these events are also not sufficient to allow increased lipolysis in the presence of TNF-$\alpha$ (*Green et al., 2004*). In those studies we showed that the effects of TNF-$\alpha$ on rates of lipolysis were dependent on the presence of glucose although the effects on perilipin and mitogen-activated protein kinase phosphorylation were independent of glucose (see Fig. 5).

We have reported that the stimulatory effect of TNF-$\alpha$ on lipolysis occurs only in the presence of glucose (*Green et al., 2004*). Therefore we determined whether this glucose

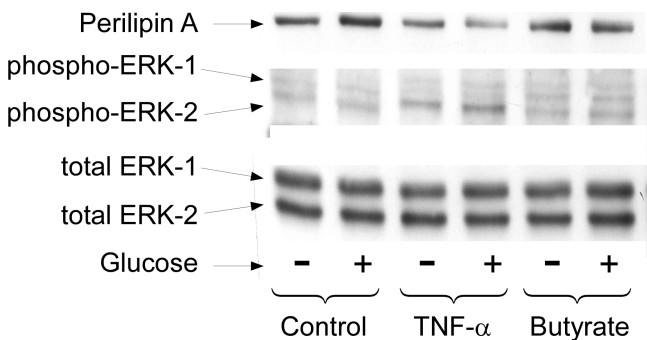

**Figure 5 Effect of butyrate on MAPK and perilipin.** 3T3-L1 adipocytes were treated with or without 25 mM glucose and 50 ng/ml TNF-α or 5 mM butyrate for 24 h. Total protein extracts were prepared and Western blots were performed with antibodies raised against (A) perilipin (approx 57 kDa), (B) phosphorylated MAP Kinase or (C) total MAP Kinase (ERK 1 and 2 ran at approx. 44 and 42 kDa, respectively). Data depicted are representative of three or more independent experiments.

dependence is true also for the lipolytic effect of butyrate. Figure 6A shows that butyrate stimulates lipolysis only when glucose is present in the incubation medium. Similarly, the lipolytic effect of the HDAC inhibitor trichostatin A occurred only in the presence of glucose (Fig. 6B). The lipolytic effect of propionate was also glucose-dependent (data not shown). Changes in energy status due to glucose deprivation are reflected in increased AMP/ATP ratios which in turn lead to phosphorylation of AMPK. Shown in Fig. 7 is a western blot of total protein extracts from cells treated for 6 h with TNF-α or butyrate in the presence or absence of glucose. As seen in Fig. 7A, phosphorylation of AMPK (T172) was decreased in cells treated with glucose compared to those without, regardless of the presence of TNF-α or butyrate. So although AMP/ATP ratios appear to be affected by short periods of glucose deprivation, phosphorylation of AMPK is dependent only on the presence of glucose and therefore cannot explain the increased rates of lipolysis, which also require the presence of TNF-α or butyrate.

To investigate whether the glucose requirement is identical for TNF-α and for butyrate we investigated the requirement for glucose metabolism using the glycolytic inhibitor iodoacetate. Glyceraldehyde-3-phosphate dehydrogenase is specifically inhibited by 100 μM iodoactate, whereas other glycolytic enzymes are inhibited by iodoacetate only at millimolar concentrations (Webb, 1966).

Iodoacetate (100 μM) prevented the increased rates of lipolysis with TNF-α and glucose (Fig. 8B) supporting our previous data that glucose metabolism is required for the glucose effect (Green et al., 2004). Iodoacetate also prevented the increased rates of lipolysis with butyrate and glucose (Fig. 8C). However, the glucose-dependence of TNF-α correlated well with the ability of glucose to be metabolized to lactate, whereas the glucose-dependence of butyrate did not. Treatment of 3T3-L1 adipocytes with TNF-α but not butyrate resulted in increased lactate release into the media (Fig. 8A). When cells were incubated in the presence of dichloroacetate (a pyruvate dehydrogenase kinase 4 inhibitor that promotes glucose oxidation over conversion to lactate) TNF-α no longer increased rates of lipolysis (Fig. 8B). In marked contrast, dichloroacetate had no effect on butyrate-stimulated

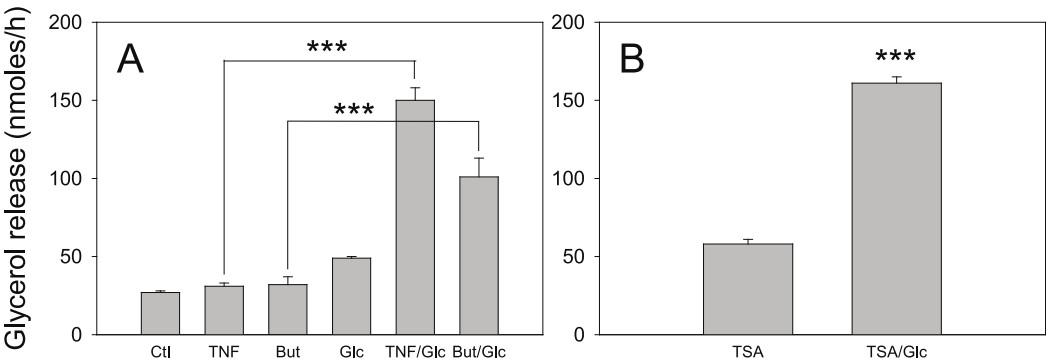

**Figure 6 Effect of glucose on the lipolytic effect of butyrate and trichostatin A.** (A) 3T3-L1 adipocytes were incubated in glucose free media supplemented with pyruvate for 24 h with no additions (Ctl), 50 ng/ml TNF-$\alpha$ (TNF), 5 mM butyrate (But), 25 mM glucose (Glc), or in combinations as indicated. Cells were washed and glycerol release measured as in the legend for Fig. 1. (B) 3T3-L1 adipocytes were treated with 1 μM Trichostatin A (TSA) in the presence or absence of 25 mM glucose. Data depicted are representative of three or more independent experiments. Data shown are means ± SE ($n = 3$). ***$P < 0.001$.

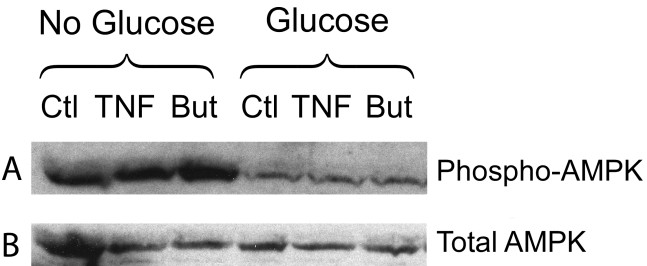

**Figure 7 Effect of butyrate on phosphorylation of AMPK.** 3T3-L1 adipocytes were treated with or without 25 mM glucose and 50 ng/ml TNF-$\alpha$ or 5 mM butyrate for 6 h. Total protein extracts were prepared and Western blots were performed with antibodies raised against phospho-AMPK (T172) (A) or total AMPK (B). Both ran at approx. 62 kDa. Data depicted are representative of three or more independent experiments.

lipolysis (Fig. 8C), suggesting that the glucose effects on TNF-$\alpha$- and butyrate-stimulated lipolysis are mechanistically distinct.

## DISCUSSION

We have demonstrated that butyrate increases the rate of glycerol release in 3T3-L1 adipocytes. This effect of butyrate was slow to develop, suggesting that changes in gene expression are involved, rather than rapid mechanisms that would affect, for example, production of a second messenger.

In a previous report we demonstrated that 3-hydroxybutrate had a small inhibitory effect on lipolysis in primary rat adipocytes, but that butyrate had no effect (*Green & Newsholme, 1979*). However, these were short-term (1-hour) experiments, and so are consistent with the present study. As far as we are aware, these are the only studies on effects of butyrate on lipolysis.

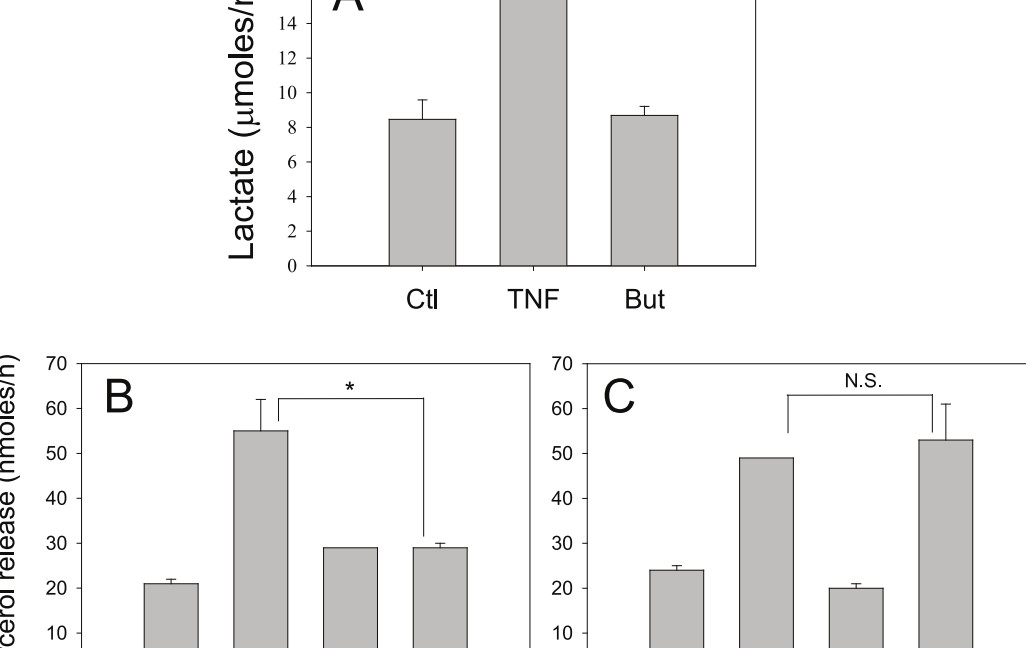

**Figure 8** **Effect of iodoacetate and dichloroacetate.** (A) 3T3-L1 adipocytes were treated in 25 mM glucose for 6 h with no additions (ctl), 50 ng/ml TNF-$\alpha$ (TNF), or 5 mM butyrate. (B) Adipocytes were treated with TNF and glucose and either 100 µM iodoacetate (Iod) or 100 µM dichloroacetate (DCA) and then glycerol release was measured as in the legend for Fig. 1. (C) Same conditions as (B), but with butyrate. Data depicted are representative of three or more independent experiments. Data shown are means $\pm$ SE ($n = 3$). *$P < 0.05$; **$P < 0.01$.

Butyrate and other HDAC inhibitors, such as trichostatin A, have been used for many years in the laboratory to enhance expression from viral promoters (*Marshall et al., 2003*; *Li et al., 1994*; *Condreay et al., 1999*). In addition, the effects of butyrate and certain other short-chain fatty acids on endogenous gene expression have been studied extensively and their effects have been attributed to their inhibition of histone deacetylase (*Kruh, 1982*; *Sealy & Chalkley, 1978*). The histone deacetylase inhibitory activity of butyrate alters $\beta$-adrenergic receptor profiles in adipocytes (*Krief et al., 1994*; *Ding et al., 2000*). Although receptor profiles were not assessed in the present study, the PKA inhibitor H-89 prevented butyrate-mediated increases in rates of lipolysis, suggesting that changes in intracellular cyclic AMP are important in the effect. The ability of H-89 to prevent butyrate-mediated increases in rates of lipolysis is in contrast to its lack of effect on serum-stimulated lipolysis in primary adipocytes (*Rumberger & Peters, 2004*).

Butyrate has been reported to affect gene expression and induce differentiation of Swiss 3T3 into adipocytes (*Toscani, Soprano & Soprano, 1990*) supporting our view that the effects we have observed are likely due to changes in gene expression. Further work will be required to confirm this and investigate the genes involved.

While acute regulation of lipolysis (e.g., by hormones such as insulin) is well documented, factors that regulate lipolysis over longer time periods are poorly understood. However, it is known that lipolysis can be regulated chronically in vivo. For example, we have reported that expression of hormone-sensitive lipase decreases in obese subjects after weight loss (*Klein et al., 1996*), and short-term fasting has been reported to increase expression of this enzyme (*Sztalryd & Kraemer, 1994*). Aging and obesity are both associated with chronic alterations in adipose tissue lipolysis (*Arner, 1999*; *Dax, Partilla & Gregerman, 1981*; *Hickner et al., 1999*).

While butyrate is established as an HDAC inhibitor, we considered various other explanations for its stimulatory effect on lipolysis. One possibility was that the mechanism is similar to that of TNF$\alpha$, one of the few other agents known to have long-term effects on lipolysis in 3T3-L1 adipocytes. However, TNF$\alpha$ has been reported to stimulate lipolysis through the ERK1/2 pathway in these cells, leading to down-regulation of perilipin (*Rosenstock, Greenberg & Rudich, 2001*; *Souza et al., 1998*; *Souza et al., 2003*; *Gronning et al., 2002*). We found similar effects of TNF$\alpha$, but not butyrate, essentially eliminating this as a mechanism for the butyrate effect.

The small molecule HDAC inhibitor, trichostatin A mimicked the effect of butyrate on lipolysis, including the glucose-dependence of the effect. We found that the lipolytic activity of other SCFA was not directly related to carbon chain length per se, but to their relative potency as HDAC inhibitors. It is important to note that at the concentrations used, butyrate and propionate, but not acetate or 2-amino butyric acid, exhibit significant HDAC inhibitory activity (*Marshall et al., 2003*; *Kruh, 1982*). Propionate (three carbons instead of four) inhibits HDAC less potently than butyrate, showed a similar response with lipolysis. The effect of propionate on lipolysis was also glucose-dependent. By contrast two closely related molecules that inhibit HDAC comparatively poorly (four-carbon 2-amino butyric acid and two-carbon acetate) did not stimulate lipolysis at concentrations where related compounds with HDAC inhibitory activity has maximal effects on lipolysis. Thus the lipolytic activity of SCFA cannot be directly attributed to carbon chain length and instead appears to be correlated with HDAC inhibitory activity.

Another possibility is that butyrate acts through a GPCR. SCFA have been shown to be ligands for the orphan G protein-coupled receptors (GPCR) GPR41 and GPR43 (*Brown et al., 2003*). However, neither the absolute concentrations we have used (i.e., millimolar) nor the order of potency of the SCFA we have observed are consistent with an effect on GPR41 or GPR43. For HDAC inhibition the order of potency is butyrate > propionate > acetate (*Marshall et al., 2003*; *Kruh, 1982*), which is consistent with effects we observed on lipolysis, whereas for GPCR activation the relative order is acetate > propionate > butyrate GPR43 (*Brown et al., 2003*)). Second, activation of GPCR is rapid (minutes or less) whereas the effect of butyrate on lipolysis was slow to develop (hours), consistent with a requirement for new protein synthesis that would be expected as a manifestation of the HDAC inhibitory activity.

The stimulatory effect of SCFA on lipolysis was dependent on the presence of glucose in the incubation medium. The requirement for glucose cannot be readily explained by

alterations in cellular energy status. First, pyruvate was present as an alternative energy source and we previously reported that cellular ATP concentrations were similar to control after 16 h glucose deprivation (*Green et al., 2004*). Second, phosphorylation of AMPK, a biosensor for increased intracellular AMP/ATP ratios, was dependent only on the presence of glucose, whereas increases in rates of lipolysis also required either butyrate or TNF-$\alpha$. That AMPK activation is not causal is consistent with a recent report that concluded that activation of AMPK in adipocytes by agents that increase cyclic AMP levels is a consequence of lipolysis and not the direct result of increases in cyclic AMP levels or PKA activity (*Gauthier et al., 2008*). Our data, previous and reported here, suggest that the primary mechanism by which TNF-$\alpha$ increases rates of lipolysis is through enhancing glucose uptake and metabolism to lactate. Indeed, both TNF-$\alpha$ and HDAC inhibitors have been shown to increase glucose uptake, but unlike TNF-$\alpha$, the HDAC inhibitors appear to be muscle specific and do not affect glucose uptake in 3T3-L1 adipocytes (*Wang, O'Brien & Brindley, 1998*; *Takigawa-Imamura et al., 2003*). These reports are consistent with our data showing that TNF-$\alpha$ but not butyrate increases release of lactate into the culture media. Although the glycolytic inhibitor iodoacetate prevented increased lipolysis with both TNF-$\alpha$ and butyrate, suggesting glucose metabolism is important, the differential effect with the pyruvate dehydrogenase kinase 4 inhibitor, dichloroacetate, suggests that the mechanism of glucose action is distinct.

In conclusion, these data demonstrate that certain SCFA, as well as trichostatin A, increase the rate of lipolysis in 3T3-L1 adipocytes. Further work will be necessary to establish a causal relationship between HDAC inhibition and lipolysis, and details of the mechanisms involved. Furthermore, it is important to emphasize that the present studies were performed in cultured cells, and that animal studies are needed to confirm the findings. Nevertheless, because HDAC inhibitors are being actively investigated as potential therapeutic agents for a number of diseases including diabetes (*Christensen et al., 2011*), it will be important to determine whether such inhibitors increase circulating free fatty acid concentrations, which would likely worsen insulin resistance and possibly have adverse effects in diabetes.

**Abbreviations**

| | |
|---|---|
| **SCFA** | short-chain fatty acids |
| **HDAC** | histone deacetylase |
| **ERK** | extracellular signal-regulated kinase |
| **AMPK** | AMP-activated protein kinase |
| **TNF-$\alpha$** | Tumor necrosis factor alpha |
| **GPCR** | G protein-coupled receptor |
| **PKA** | protein kinase A |
| **H-89** | N-(2-(p-Bromocinnamylamino)ethyl)-5-isoquinolinesulfonamide dihydrochloride |

### Funding

This work was supported by the Stephen C. Clark fund. The funders had no role in study design, data collection and analysis, decision to publish, or preparation of the manuscript.

### Grant Disclosures

The following grant information was disclosed by the authors:
Stephen C. Clark fund.

### Competing Interests

John M. Rumberger is an employee of Bassett Healthcare.

### Author Contributions

- John M. Rumberger and Allan Green conceived and designed the experiments, performed the experiments, analyzed the data, contributed reagents/materials/analysis tools, wrote the paper, prepared figures and/or tables, reviewed drafts of the paper.
- Jonathan R.S. Arch conceived and designed the experiments, analyzed the data, wrote the paper, reviewed drafts of the paper.

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
