# Peer review of "Butyrate and other short-chain fatty acids increase the rate of lipolysis in 3T3-L1 adipocytes"

_PeerJ, doi:10.7717/peerj.611_

## Round 0.1 · original submission · Major Revisions

As you will see, the comments of the reviewers are quite diverse, going from minor revision to straight rejection because of lack of novelty. I'd, therefore, strongly suggest you to try to answer all these comments (and especially the comment about the lack of novelty and the fact that studies were only made on 3T3 cells). The ball is clearly in your camp.

·

Basic reporting

This manuscript has investigated how lipolytic activity is influenced in 3T3-L1 adipocytes through a series of treatment rationales including. This work is important as developing therapies for diabetes are utilising histone deactylase inhibitors therefore insight into the intracellular mechanism and action of these pathways as noted in this manuscript is important and relevant for clinical understanding. This current paper has specifically sought to evaluate the influence of butyrate and short chain fatty acids effects on lipolysis. These studies have highlighted that stimulation of lipolysis by butyrate was glucose dependent, with it also noted that TNF-alpha increases lipolysis through glucose uptake and metabolism of lactate. In addition glycolytic inhibitors prevented increased lipolysis in the presence of either TNF-alphaor butyrate. However the mechanism of lipolysis action appears different between TNF-alpha and butyrate. Further intracellular evaluation noted the importance of short chain FAs to increased lipolysis due to the inhibition of histone deactylase (HDAC), whilst neither GPCR, MAPK nor AMPK were involved. Taken together this work extends our current knowledge and importance of how HADC can be influenced by butyrate and SCFA, representing an significant and interesting body of work in the field.

Experimental design

Page 5 line 97: Could the authors cite one of their previous references for this methodology and repeat for the other methods used as deemed appropriate.

Figure 5 & 7: Could the authors add the KDa sizes on the western blots

Validity of the findings

No comments

Additional comments

Minor corrections

Page 5 line 98: remove second full stop
Page 7 line 136: replace ‘Because’ with ‘As’
Page 10 line 217: place e.g. in italics
Page 13: line 277; the full stop missing at the end of the sentence

·

Basic reporting

1. The corresponding author of this manuscript, Dr. Allan Green, had previously reported that 3-hydroxybutyrate inhibits basal lipolysis and enhances the anti-lipolytic action of insulin in rat primary adipocytes. Also, the author reported that butyrate had little effect on lipolysis of rat adipocytes (Allan Green and Eric. Newsholme. Biochem J. 1979; 180: 365-370). This seems the only study investigating the lipolytic effect of butyrate, and it was published by the author of the current manuscript, so the relevant information should be introduced or discussed.

2. Also, Toscani et al. (J Biol Chem, 1990; 265: 5722-30.) reported that butyrate induces adipocyte differentiation and examined the effects of butyrate on expression of several adipogenic genes, by the reliable RNase-protection assay. The present manuscript assumes that butyrate action might be relevant to gene expression; the above reference may be discussed.


3. Citation: Page 8: Long term regulation of lipolysis by other mediators, such as TNF-α, is thought to involve activation of the MAP kinases ERK1&2 and the down-regulation of perilipin [22;23]. Page 10, TNFα has been reported to stimulate lipolysis through the ERK1/2 pathway in these cells, leading to down-regulation of perilipin [34;35]. The authors discussed exactly the same thing, but cited different references.

4. Usually, statistical probability (p) may be expressed in a pattern as *p < 0.05, **p < 0.01, and ***p < 0.001. The use of *, **, *** marker is confusing in this manuscript, for example: Figure 6: * P<0.0001; ** P<0.001; Figure 2: **P<0.05; * P<0.0001; Figure 8: * P<0.05; ** P<0.003.

5. Abstract: ' unlike TNF- - ,' butyrate-stimulated lipolysis. Lane 98: "and then incubated for another 1 hour..", there is 2 stops.

6. PeerJ uses the "Name. Year" style in-text citation

Experimental design

Page 6 and Figure 1, Importantly!
At 1, 2, 3, 4 hours after treatment, the 3T3-L1 adipocytes were washed and incubated for another 1 hour, and then this 1-h glycerol release/accumulation was measured as lipolytic index. The control adipocytes without treatment at the 3- and 4-hour time points, show significantly decreased level of glycerol release. Such decrease is a surprise. Without treatment, groups (1, 2, 3, 4 hours) of the control adipocytes should have a comparable level of lipolysis. This issue is important for evaluating the conclusion that butyrate increases lipolysis.

Fully differentiated, matured 3T3-L1 adipocytes tend to float up or the edge of the adipocyte layer might be easily folding, particularly when they are washed lately at 3-10 days post differentiations. If these happened, glycerol release might be decreased because of insufficient number of adipocytes. Another concern is the normalization of the lipolysis. The authors only normalized the glycerol release to per well cell (nmol per hour per 24-well). This is ok but there is a common problem that the differentiation of 3T3-L1 preadipocytes may often vary obviously from well to well. One might seek alternative approach, such as normalization to cellular protein content, cell number (DNA content), or oil content.

Validity of the findings

Catecholamine stimulate lipolysis within minutes. Butyrate induces lipolysis within 1~2 hours, the time period is not so slower. Also, butyrate treatment increases cellular cAMP level, a typical lipolytic signaling in adipocytes. It is stated that changes in gene expression and new protein synthesis might be involved in the lipolysis of butyrate (within few hours??). However, the study fails to show when and what genes or proteins is changed upon butyrate stimulation.

Results: Page 6, first paragraph --- It was stated that "After 1 h treatment with butyrate the rate of lipolysis was similar to that of control cells". However, the text description is unmatched with the data of Figure 1, which shows p<0.05 at the 1-hour point.

Because butyrate is a short fatty acid present at high level in the body, the finding that butyrate increases lipolysis in differentiated 3T3-L1 adipocytes is particularly interesting. It would largely strengthen the work if the lipolytic action of butyrate is confirmed in authentic model of adipose cell, such as isolated primary adipocytes or adipose tissue fragments.

Reviewer 3 ·

Basic reporting

Nothing is really new in this article. Furthermore, experiments are performed only in 3T3 cells. Does authors have datas in animal model ?

References list isn't fully exhaustive.

Experimental design

No comments

Validity of the findings

I think there is no new knowledges providing here

---

## Round 0.2 · Minor Revisions

I do apologize for the long delay it took to come with this decision, but as you may have guessed, I was facing the quite contradictory reviewers' reports. I, therefore (i) made a complete analysis of the revised paper my-self; (ii) seek additional advice. Based on this, I'm happy to let you know that the paper, in its revised form, is acceptable for publication if you agree to make the additional changes indicated hereunder.

1. Please, mention in the Discussion (preferably close of in the conclusions) that your data were obtained with cultured cells, which means that further animal studies are needed before you can state, as you do in your submission, that "...because HDAC inhibitors are being actively investigated as potential therapeutic agents for a number of diseases including diabetes (Christensen et al., 2011), it will be important to determine whether such inhibitors increase circulating free fatty acid concentrations, which would likely worsen insulin resistance and possibly have adverse effects in diabetes.". This sentence, indeed, has a very strong connotation to clinical applications. Your study in cultured cell is insufficient to make suggestions about potential clinical applications.

2. Figure 1: there is no data for time 0 here. Assuming these would be similar to the tile 1h for control, we ween that the production of glycerol in cells incubated wit butyrate increase almost linearly with time over the first 3 hours. Your text does not really convey this information and may even be misleading since you suggest that nothing happens during the first hour, which is probably incorrect. This was mentioned by one of the reviewer, but you decided to ignore it. I guess you should modify your wording and be more cautious about your conclusions in this context.

3. Please, correct the manuscript as per the suggestion #4 of reviewer #2 (which you forgot and, therefore,incorrectly, numbered his/her comments in your rebuttal. This comment had to do with the confusion created by different symbols used for statistical significance (sometimes * is p<0.000 [Fig. 2] but sometimes its is < 0.05 (Fig. 1). All that is creating confusion. Please, recheck your manuscript for consistency in this context.

4. There are (too) many acronyms in your text. Please, consider what is in the Instructions to Authors of several high-level Journals and that I paraphrase here: "Abbreviations should be used as an aid to the reader, rather than as a convenience to the author, and therefore their use should be limited". Please, recheck you text and make sure that acronyms are only introduced used if the corresponding words are used at least 5 times in the paper.

Please, submit your corrections and/or your rebuttal whenever possible.

---

## Round 0.3 · accepted · Accept

I guess that the second round of review resulted in a stronger paper while being more cautious about the clinical significance of the data.